# Prediction of Asbestos-Related Diseases (ARDs) and Chrysotile Asbestos Exposure Concentrations in Asbestos-Cement (AC) Manufacturing Factories in Zimbabwe

**DOI:** 10.3390/ijerph20010058

**Published:** 2022-12-21

**Authors:** Benjamin Mutetwa, Dingani Moyo, Derk Brouwer

**Affiliations:** 1School of Public Health, Faculty of Health Sciences, University of the Witwatersrand, Johannesburg 2193, South Africa; 2Faculty of Medicine and Health Sciences, Midland State University, Gweru 054, Zimbabwe; 3Department of Community Medicine, Faculty of Medicine, National University of Science and Technology, Bulawayo 029, Zimbabwe

**Keywords:** prediction, asbestos related diseases, exposure concentration, cumulative exposure, chrysotile, Zimbabwe

## Abstract

The use of historical asbestos measurement data in occupational exposure assessment is essential as it allows more quantitative analysis of possible exposure response relationships in asbestos-related disease (ARD) occurrence. The aim of this study was to predict possible ARDs, namely lung cancer, mesothelioma, gastrointestinal cancer, and asbestosis, in two chrysotile asbestos cement (AC) manufacturing factories. Prediction of ARDs was done using a specific designed job-exposure matrix for airborne chrysotile asbestos fibre concentrations obtained from the Harare and Bulawayo AC factories and through application of OSHA’s linear dose effect model in which ARDs were estimated through extrapolation at 1, 10, 20, and 25 years of exposure. The results show that more cancer and asbestosis cases are likely to be experienced among those exposed before 2008 as exposure levels and subsequently cumulative exposure were generally much higher than those experienced after 2008. After a possible exposure period of 25 years, overall cancer cases predicted in the Harare factory were 325 cases per 100,000 workers, while for the Bulawayo factory, 347 cancer cases per 100,000 workers exposed may be experienced. Possible high numbers of ARDs are likely to be associated with specific tasks/job titles, e.g., saw cutting, kollergang, fettling table, ground hard waste, and possibly pipe-making operations, as cumulative exposures, though lower than reported in other studies, may present higher risk of health impairment. The study gives insights into possible ARDs, namely lung cancer, mesothelioma, gastrointestinal cancer, and asbestosis, that may be anticipated at various cumulative exposures over 1, 10, 20, and 25 years of exposure in AC manufacturing factories in Zimbabwe. Additionally, results from the study can also form a basis for more in-depth assessment of asbestos cancer morbidity studies in the AC manufacturing industries.

## 1. Introduction

Asbestos in all its forms (serpentine group—chrysotile, amphibole group—crocidolite, amosite, tremolite, actinolite, and anthophyllite) is an important occupational carcinogen causing about half of cancer-related deaths [1,2]. There is sufficient evidence in humans for the carcinogenicity of all forms of asbestos from which the International Agency on Research on Cancer (IARC) concluded that asbestos causes mesothelioma and cancer of the lungs, larynx, and ovaries and that asbestos is a Group 1 carcinogen [2]. Additionally, a positive association between exposure to all forms of asbestos and cancer of the pharynx, stomach, and colorectum has been established. However, with respect to the colorectum, the IARC Working Group was evenly divided to conclude that evidence for asbestos was sufficient enough to be considered as causing cancer. Furthermore, experimental animal studies have demonstrated that there is sufficient evidence showing carcinogenicity of all forms of asbestos [2]. Gastrointestinal tract cancers have also been reported in groups of persons occupationally exposed to amosite, chrysotile, or mixed fibres containing chrysotile [2]. All forms of asbestos cause asbestosis [1,2,3].

The latency period associated with these diseases can range from 10 to 40 years from exposure [1,2,4]. However, short intense exposures to asbestos, lasting from several months to 1 year or more, can be sufficient to cause asbestosis [4]. The health risks of exposure to all forms of asbestos are largely associated with inhalation [2]. Studies of workers exposed to chrysotile asbestos fibre in different sectors have broadly demonstrated exposure–response or exposure–effect relationships for chrysotile-induced asbestosis, as levels of exposure have increased and hence, such exposures have been reported to result in increased incidence and severity of disease [3]. Asbestosis stage changes have been noted to be common following prolonged exposure of 5 to 20 f/mL [1].

Use of measurement data in occupational exposure assessment is essential as it allows more quantitative analysis of possible exposure response relationships [5]. Furthermore, historical exposure to asbestos is an important factor in asbestos-related disease occurrence because asbestos materials have been used for the past few decades in many workplace settings [6,7]. In Zimbabwe, exposure to chrysotile asbestos has been ongoing for many decades since the mines were opened around 1910 and manufacturing began around 1943 [8]. In occupational exposure assessment studies, cumulative exposure is often used as an exposure metric in quantitative epidemiologic evaluation studies [9,10,11,12,13,14,15,16,17]. Cumulative exposure is normally defined in terms of fibre/millilitre years (f/mL-years) and the definition is based on the level of exposure in the workplace, measured as the number of fibres found in each ml of air, in the air which a worker breathes at work and multiplied by the number of years or fraction of a year worked at that level [18,19,20,21].

Early studies have reported that a worker exposed to 100 f/mL-years (for example, 50 years of exposure at 2 f/mL or 25 years at 4 f/mL or 10 years at 10 f/mL) are likely to have a 1% chance of developing asbestosis [22]. Thus, the correlation of exposure concentrations with disease occurrence among persons exposed provided a basis for setting an occupational exposure limit for asbestos fibres in the ambient air [18,19]. Since accurate quantitative exposure concentration data are sometimes difficult to obtain, cumulative exposure expressed as fibre/mL-years is usually used as a common metric in exposure assessments reported in epidemiological studies [18,19,23]. Additionally, it has been noted that the variable cumulative exposure assumes that the duration of exposure and exposure concentration carry the same weight [18,19]. Furthermore, exposure at low concentration levels over a long period of time may mathematically be equivalent to exposure at high concentration for short periods; however, the biological effects may be different. Hence, duration or concentration levels maybe more important than their product in predicting occurrence of disease. In the case of mesothelioma, duration of exposure has been reported to appear as the most important factor in occurrence of mesothelioma [16,18,19].

In a study on quantitative risks of mesothelioma and lung cancer in relation to asbestos exposure, it has been reported that a cumulative exposure of 1 f/mL-years for crocidolite yields a lifetime risk for mesothelioma of 650/100,000 of exposed persons. Similarly, the estimates for amosite and for chrysotile were 90/100,000 and 5/100,000, respectively [14]. The authors concluded that at exposure levels seen in occupational cohorts, the exposure specific risk of mesothelioma from the three principal commercial asbestos types is largely in the ratio of 500:100:1 for crocidolite, amosite, and chrysotile, respectively, and that the risk differential between chrysotile and amphiboles (crocidolite and amosite) with respect to lung cancer was somewhere between 1:10 and 1:50 [14]. However, the study by Hodgson and Darnton was observed to have limitations in that the meta-analysis carried out was based on studies for which the quality of the cohort studies used were not considered and that the range of uncertainty in risks of lung cancer and mesothelioma was wide [24,25]. In a study of textile workers from four plants in North Carolina, USA, in which workers were primarily exposed to chrysotile, Loomis and co-authors reported an overall excess of lung cancer, mesothelioma, and pleural cancer amongst the workers, pointing to the potency of chrysotile asbestos [26,27].

A reduction of exposure to 0.2 f/mL of chrysotile asbestos fibre concentration was reported to result in a lifetime incidence of asbestosis of about 0.5%. Furthermore, by reducing the occupational exposure limit from 2 f/mL to 0.1 f/mL, the risk of cancer mortality was observed to be reduced by 95% from estimated cases, as high as 6411 to 336 deaths per 100,000 workers [18], over a 45-year period of exposure. On the other hand, the incidence of asbestosis was calculated to be 250 cases per 100,000 workers after exposure to asbestos at 0.1 f/mL [18].

Prediction of ARDs can be done through models that use direct or indirect estimates of asbestos exposure [28]. Direct estimates make use of exposed persons where airborne asbestos fibre levels are measured over time while indirect estimates use information about total or fibre-specific imports [28].

The risk associated with exposure to asbestos has been reported to generally follow a linear model wherein relative risk of asbestos disease is linear in dose [23,29,30,31]. The US Occupational Safety and Health Administration (OSHA) selected a linear model to describe the relationship between the excess relative risk of lung cancer and asbestos exposure [22]. Evidence of the linear dose-response relationship for lung cancer has been observed for several studies with respect to cumulative asbestos exposure in the workplace [18,19,23,32,33,34,35]. Additionally, a study by Finkelstein observed that the rate of deaths from mesotheliomas were proportional to the magnitude of cumulative asbestos exposure and that the exposure response data set was linear in dose. Furthermore, Berry et al. and Finkelstein have demonstrated an approximate linear relationship between asbestosis incidences and cumulative exposure [22,36]. These studies and many others upon which OSHA based its risk assessment models to predict asbestos-related cancers demonstrated linear relationships over a range of observations.

The approach using the OSHA Risk Assessment Model as initially conceived with respect to the objective of the study was used with the assumption that a linear relationship possibly exists between exposure and effect. Furthermore, use of the linear design in the dose-response relationship has also been assumed and applied in this study as it presented some advantages, namely:Point estimates (average exposures) can be made without knowledge of the individual exposures in the groups, suggesting that excess mortality of an entire exposed group can be related to the average exposure of the group [30].Extrapolation to various exposure circumstances can be made easily.It is likely to be a conservative extrapolation in the context of human health [30].

In this study, the direct method was applied to predict asbestos-related mortality cases, namely lung cancer, mesothelioma, and gastrointestinal cancer as well as asbestosis in two AC manufacturing factories in Zimbabwe, as airborne chrysotile asbestos fibre concentrations over the period of 1996 to 2020, reported in a previous study by Mutetwa et al. [37] and captured in a job-exposure matrix, were available [38].

## 2. Materials and Methods

### 2.1. The Factories

The study involved two major AC manufacturing factories situated in Harare and Bulawayo. Activities in the two factories included the manufacture of chrysotile cement sheets, facia boards, garden ware, and AC pipes in the case of the Bulawayo factory. The factories were built in the 1940s and 1950s and have been using chrysotile asbestos in the manufacture of AC products since their establishment. Estimates from the factories suggest that between 1996 and 2020, the estimated number of workers ranged from about 390 to 420, with the Harare factory possibly having 156 to 220 workers and Bulawayo factory having 190 to 204 during this period. Within the operational areas studied, approximately more than 100 and 150 workers in Harare and Bulawayo factories, respectively, were in various jobs as indicated in Appendix A. The jobs used in predicting ARDs have been described previously by Mutetwa et al. [37,38]. The main raw materials used in the manufacture of asbestos cement products were cement, water, and chrysotile asbestos. Generally, the AC products contain 10–20% asbestos and 80–90% cement and water.

### 2.2. Measurement of Airborne Chrysotile Fibre

Measurement of airborne chrysotile asbestos fibres was done by the company’s safety and health department using the standard method of the Asbestos International Association (AIA) Reference method for the determination of airborne asbestos fibre concentrations at workplaces by phase contrast light microscopy. In summary, a personal sampling pump set at 1 L/min flowrate was connected to a sampling train, consisting of plastic tubing and a sample holder (cowl) with a 25 mm membrane filter. The whole sampling train of the pump, tubing, sample holder, and filter was hooked to a worker. The pump was then switched on and sampling took place over a period of about four hours, after which the filters were removed, placed at the appropriate labeled slides, and treated with acetone vapour to clear. Using a hypodermic syringe, a drop of triacetin was placed onto the acetone-cleared filters and covered with a cover slip. The treated filters on the slides were stored for 24 h, after which counting of the fibres took place using a phase contrast microscope. The fibres counted were generally longer than 5 µm with a width of less than 3 µm and length-to-width ratio of more than 3:1 [37].

### 2.3. Chrysotile Exposure Concentration Data

Personal exposure chrysotile concentration data measured for the period of 1996 to 2020 extracted from paper records of the two main AC manufacturing factories in Harare and Bulawayo cities were used to assess possible estimates of ARDs, namely lung cancer, mesothelioma, gastrointestinal cancer, and asbestosis in the AC manufacturing factories. As reported by Mutetwa et al. [38], a total of 3066 personal airborne chrysotile measurements collected from company records, spanning a period of about 25 years from which 1788 monthly mean chrysotile personal exposure concentrations were drawn, constituted the data set. In combination with the US Occupational Safety and Health Administration (OSHA) linear dose–response relationship model, chrysotile personal exposure concentrations were used to generate estimates of cancer mortality cases and asbestosis cases.

### 2.4. Application of OSHA’s Risk Assessment Models

Table 1 below, reproduced from OSHA linear dose risk assessment models, provides insight into estimates of asbestos cancer mortality per 100,000 exposed workers [18,19,22].

For lung cancer, mesothelioma, gastrointestinal cancer, and asbestosis, OSHA generally relied on a relative risk model that was linear in cumulative exposure/dose. Linear regression equations established for lung cancer, mesothelioma, and gastrointestinal cancer by plotting estimates of cancer mortality cases versus respective cumulative exposures (Figure 1, Figure 2 and Figure 3) were applied to estimate possible cancer mortality cases that may arise due to exposure levels depicted in Table 2.

For asbestosis, the linear cumulative dose equation, Ra = m(f)(d), where Ra-predicted incidence of asbestosis, m-slope of linear regression taken as 0.055, f-asbestos fibre concentration, and d—duration of exposure [18,19], was used to estimate possible asbestosis cases over the respective duration of exposure.

Additionally, it has been assumed that at the average exposure concentrations indicated for respective time periods in Table 2 and further assuming that such average exposures may be experienced for durations of 1, 10, 20, and 25 years, cancer mortality cases might be observed after such duration of exposures. Exposure data collected spanned about 25 years, hence the exposure duration capped at 25 years. Furthermore, Table 2 below, adapted from the study by Mutetwa et. [38], was also used to predict ARDs by drawing exposure concentrations from various time periods for various jobs in the two AC manufacturing factories.

### 2.5. Data Analysis

Data analysis was conducted using IBM SPSS version 26. Mean personal exposure concentrations per time period and by key jobs in a particular factory location as described in Table 1 by Mutetwa et al. [38] were used to derive cancer mortality cases after exposure duration periods of 1, 10, 20, and 25 years as well as possible estimates of asbestosis cases after workers were exposed for a period of 25 years. The arithmetic mean was used as a representative value for analysis of the measurements as this is normally taken as the best summary measure of exposure in epidemiological studies of chronic diseases when adopting a linear exposure response model [6,39].

## 3. Results

Following the application of the linear regression equations derived from Table 1 as a result of the OSHA cancer risk assessment model for asbestos occupational exposure to the ambient chrysotile asbestos concentration data and cumulative exposures therein gathered from the AC manufacturing factories, Table 3 shows a summary of overall predicted cancer mortality cases (lung cancer, mesothelioma, and gastrointestinal cancer) as well as incidence rates which may be experienced after being exposed at the mean exposure levels associated with each respective time period for each respective job. In the Appendix A (lung cancer), Appendix A (mesothelioma), and Appendix A (gastrointestinal cancer) show the estimates for the specific asbestos-related cancer mortality cases predicted after 1, 10, 20, and 25 years of exposure, respectively, and by time period.

Appendix A show that the incidence of lung cancer, mesothelioma, and gastrointestinal cancer after 1 year of exposure is very low, ranging from as low as 0.0003 to 0.01% in almost all jobs if it is assumed that workers were to be exposed at various exposure levels associated with particular time periods. Furthermore, in tandem with increased cumulative exposure after 20 years of exposure, predicted cancer mortality cases show increased occurrence of asbestos-related cancers if it is assumed that exposed workers were to work for 20 years or more at levels obtained at each respective time period. As reported by Mutetwa et al. [37,38], for the Harare factory, exposure levels show a decreasing trend from 1996 to 2016; however, from 2018 to 2020, exposure levels show an upward trend, with the results that predicted cancer cases for the period of 2018 to 2020 in the Harare factory being higher compared to the preceding time period of 1996 to 2016.

For the Bulawayo factory, nonetheless with exposure concentrations showing a downward trend from 1996 to 2020, there are decreasing levels shown of predicted cancer (i.e., lung cancer, mesothelioma, and gastrointestinal cancer) mortality cases if exposed workers in various jobs work at the obtaining exposure levels associated with the respective time periods.

The results in Table 3 show that more cancer cases are likely to be experienced among those exposed before 2008, as exposure levels and subsequently cumulative exposure were generally much higher than those experienced after 2008.

Table 4 shows estimates of possible asbestosis incidences likely to be observed at various cumulative exposures after a 25-year duration of exposure and by job and time period. Cumulative exposure levels range from 1.0 to 4.8 f/mL-years, with the highest cumulative exposures being exhibited at the saw cutting operations in both factories, followed by ground hard waste operations particularly before 2008. Kollergang operations in both factories, laundry operations (Harare factory), and pipe sections operations before 2008 also exhibit relatively high cumulative exposures with concomitant increased predicted asbestosis cases. Asbestosis cases likely to be detected after a 25-year duration of exposure range from 50 to 260 cases per 100,000 (0.05% to 0.26% incidence of asbestosis) exposed workers for various jobs. Overall, on average in both factories, asbestosis cases likely to be detected were within the range of 150–170 per 100,000 workers exposed (0.15% to 0.17% incidence) to asbestosis.

Appendix A, showing overall summary estimates of cancer mortality cases by factory and duration of exposure, suggest that 15, 139, 278, and 347 cases per 100,000 workers (saw cutting, fettling table, ground hard waste, and laundry operators, respectively) of asbestos-related cancers may be experienced after 1, 10, 20, and 25 years of exposure, respectively, in the Harare factory. The Bulawayo factory with similar or the same order of magnitude of cumulative exposure levels to those obtained in the Harare factory also displays a similar trend wherein the same or similar number of cancer cases per 100,000 workers exposed are likely to be experienced at saw cutting operators, fettling table operators, as well as at pipe section after 1, 10, 20, and 25 years of exposure.

## 4. Discussion

The study attempts to predict possible ARDs in the chrysotile AC manufacturing factories in Zimbabwe by applying the OSHA linear dose–response relationship (OSHA cancer risk assessment model) for lung cancer, mesothelioma, gastrointestinal cancer, and asbestosis. The predicted ARDs drawn through the application of OSHA’s cancer risk assessment tool [18.19,22] provided possible estimates of cancer mortality cases for workers exposed to chrysotile asbestos fibres at various cumulative exposures if exposed at various exposure concentrations obtained at various time periods.

The results suggest that saw cutting operators followed by kollergang and ground hard waste operators in both factories may be at an increased risk of developing ARDs, as results (Table 3 and Table 4) show a high number of predicted cancer mortality cases and asbestosis cases especially if workers were to be exposed before 2008, where exposure concentrations and subsequently cumulative exposures were high compared to exposure concentrations for the time period of 2009 to 2020, if duration of exposure of 20 years or 25 years is applied.

The results also suggest that over a possible exposure period of 25 years, high exposure concentrations experienced before 2008 and, subsequently, cumulative exposures may possibly yield high asbestos-related cancers and asbestosis as reflected in Appendix A.

The predicted cancer cases reported in this study for 1-, 10-, 20- and 25-year durations of exposure are much lower compared to those reported by Jafari et al. [40], who also applied the OSHA linear dose response, in which there were 499 cases per 100,000 workers after 1 year of exposure and 6965 cases per 100,000 workers after 20 years of exposure [35]; and are also lower than those reported by Magnani and Leporati [41]. The high cancer cases as reported by Jafari et al. reflect high cumulative exposures at 1- and 20-year durations of exposure compared to those obtained in this study. It is insightful to note that the cumulative exposures in this study are quite low compared to cumulative exposures recorded in the 1970s where exposures as high as 200–300 f/mL-years were recorded [37], compared to maximum cumulative exposures of 4.3 to 4.8 f/mL-years for an exposure duration as high as 25 years (Appendix A) recorded in our study. Such high cumulative exposures in the past would suggest high levels of asbestos-related cancers and asbestosis compared to those predicted in this study.

Cases of ARDs in Zimbabwe have been reported mainly emanating from the mines and mills. Twenty-seven (27) had ARDs, 21 individuals had evidence of asbestosis with one related to asbestos cement manufacturing, while 3 had possible mesothelioma, and 3 possibly had lung cancer/non-malignant pleural diseases. The authors further indicated that the results presented some limitations as they reported that there was under-recognition bias introduced by looking at workers’ compensated derived cases, with the health system having no capacity to follow-up workers who quit or retired from the chrysotile asbestos industry [8]. The levels of ARDs predicted in the study by Cullen and Baloyi may be difficult to infer with respect to our study as the cases were largely from the mines and mills.

On average, over a 25-year duration of exposure, 150 and 170 cases (0.15–0.17% incidence) of asbestosis per 100,000 workers in the Harare and Bulawayo factory, respectively, may develop asbestosis at overall cumulative exposures of 2.8 f/mL-years for Harare and 3.0 f/mL-years for the Bulawayo factory.

For the time period of 1996 to 2000, the period that presents the highest exposure concentrations and subsequently high cumulative exposures, saw cutting, kollergang, fettling table, ground hard waste, and possibly pipe making operational areas may experience the highest cancer incidence rates after 20 years and possibly after a period of 25 years in both factories. Indeed, based on the estimated number of workers in the various jobs as illustrated in Appendix A, and in particular considering just the jobs with the possible highest exposures, cancer incidence cases which may be experienced following exposures associated with the time period of 1996 to 2000, may be 0.24% (saw cutting operators), 0.04% (kollergang operators), and 0.02% (ground hard waste) for the Harare factory after 20 years of exposure. After 25 years of exposure, the same operational areas of saw cutting, kollergang, and ground hard waste, based on the estimated number of workers in the various jobs, also exhibited high cancer incidence rates of 0.30%, 0.05, and 0.03%, respectively, compared to other operational areas. The Bulawayo factory follows a similar pattern for the three operational areas, with saw cutting recording a high incidence rate of 0.27% compared to other operational areas in this factory after 20 years of exposure. Additionally, as a result of high exposure concentrations and, in turn, high cumulative exposure after 25 years, the three operational areas may experience 0.14%, 0.02%, and 0.01% asbestosis cases among saw cutting, kollergang, and ground hard waste operators, respectively, in the Harare and Bulawayo factories. Given the estimated maximum total number of workers with the job title of saw cutting over both the factories (Appendix A) and exposure levels experienced by saw cutting operators, it cannot be excluded that cancer cases may have occurred.

Asbestos-related cancers derived from this study may possibly need to be taken in the context of what may constitute a significant risk to health. OSHA [23] has considered that the risk of cancer mortality of more than 1 case per 1000 workers from occupational causes presents a significant risk. Hence, from Appendix A, which depicts overall possible cancer cases over a 25-year duration of exposure, it may be deduced that the cumulative exposures in both factories, though seemingly low compared to past exposures reported elsewhere [19,42], may possibly present some significant health risk if evaluated against the OSHA significant risk criteria, as overall cancer mortality cases ranged from 2.89 to 3.47 cases in the case of Harare factory operational areas while for the Bulawayo factory, risk of cancer mortality ranged from 2.89 to 4.63 cases per 1000 workers exposed. However, if consideration is given to other risk levels such as four cases per 1000 workers exposed (The Netherlands and Germany) [43,44], then the risk range obtained above for this study may not necessarily be significant. The cancer cases in this study, per 1000 appear elevated as a result of high chrysotile exposures experienced in the earlier time period of 1996 to 2008. Moreover, as elaborated by the OSHA, while some significant risk remains at the OEL of 0.1 f/mL, health risk may also be significantly reduced at or below the OEL of 0.1 f/mL compared to exposures obtaining in the earlier years of the 1960s, 1970s, and 1980s and early 1990s in which exposures could range from 0.2–0.5 f/mL or even higher. Additionally, the risk of asbestos-related cancer may also be further reduced by reducing exposure levels as indicted in Appendix A, particularly for the Bulawayo factory, in which exposure concentrations over the years from 1996 to 2020 have been on a downward trend to the extent that exposures within the range of 0.04 to 0.07 f/mL were being realised during the time period of 2018 to 2020. This may suggest that cancer cases may possibly approach one case or less per 1000 workers as exposure concentrations decrease and if all possible control measures are implemented to the fullest extent possible. Moreover, studies on cancer mortality have demonstrated that mortality estimates from asbestos-related cancers of any type decrease significantly when exposure is reduced [18,19]. For instance, during the time period of 2009 to 2016 in the Harare factory, at exposure concentrations of 0.05 to 0.07 f/mL depending on the job, 0.9–1.3 lung cancer cases, 0.5–0.7 mesothelioma cases, and 0.09–0.13 gastrointestinal cancers per 1000 workers exposed may be experienced. Similarly for the period of 2018 to 2020, in the case of the Bulawayo factory in which exposure concentrations ranged from 0.04 to 0.06 f/mL over an exposure period of 25 years, 0.7 to 1.1 lung cancer cases, 0.4 to 0.6 mesothelioma cases, and about 0.07 to 0.11 gastrointestinal cases per 1000 workers may also be realised, taking into account exposure concentrations and cumulative exposures reflected in Appendix A for the time period of 2009 to 2020. These risk rates may possibly be low if evaluated against the four cases in 1000 workers exposed taken with respect to Netherlands or Germany as acceptable excess cancer risk levels. Zimbabwe does not have a benchmark or reference point for excess cancer risk and might as well consider the four cases in 1000 workers as starting point to consider as part of occupational cancer management policy. Furthermore, health risks presented by exposure concentrations reflected in this study may be minimised further or reduced by effective use of engineering controls, good occupational hygiene practices, and use of appropriate respiratory protective equipment and overall sustained implementation of an occupational safety and health management system across all areas of AC manufacturing processes.

ARDs predicted in this study are based on a fairly large personal airborne chrysotile fibre concentration data set spanning a period of about 25 years and thus provides a reasonable measure of cumulative exposures and possible relationship with diseases which may be associated with various jobs. Cumulative asbestos exposures obtained in this study may be considered as low and hence may provide a good basis for further studies on chrysotile exposure–response relationships relating to asbestos cancer morbidity or mortality cases in Zimbabwe. The study further provides unique insights into the possible estimates of ARDs as it relates to lung cancer if a chrysotile exposure mortality and or morbidity study is carried out in the future, more so that cumulative exposures may generally give robust answers to the existence of an association, regardless of the underlying true mechanism of disease. Although the OSHA risk assessment tool or dose response relationship model shows that excess risk is linear in dose, excess cancer mortality or morbidity cases may not really be linear especially at high exposure levels. Another limitation of the study was that the possible number of ARDs depicted in this study may also not really reflect the true ARDs cases, as the OSHA risk assessment was also based on studies that included a significant amount of amphibole asbestos, which are generally considered to have a higher potency than chrysotile asbestos [14,29], while chrysotile asbestos used in the AC manufacturing industries in Zimbabwe has been mainly chrysotile, and thus the disease experience which is exhibited in the AC manufacturing factories in Zimbabwe may be different from workplace settings associated with studies used in the development of the OSHA risk assessment model. Furthermore, the epidemiological data used in developing a linear dose relationship for human exposure to asbestos are limited as a result of the fact that current health effects are largely a result of past exposures when exposure controls were inadequate and exposure measurements were imperfect, assuming a number of scenarios were in some cases conversion factors with uncertainties that were applied in expressing asbestos fibre concentrations in f/mL. The dose response data with respect to gastrointestinal cancer have also been limited compared to those for lung cancer and mesothelioma.

Limitations of the OSHA risk assessment approach as applied in this study also include uncertainty in risk by extrapolation from high occupational exposure levels to much lower levels, mass to fibre conversion factors used in modelling the linear in dose response models, and variability in exposure estimates that are built into the linear dose models [30]. Despite these limitations, the personal exposure data used in this study and applied in the context of the OSHA risk assessment linear dose relationship, provide some possible estimates of asbestos disease under the exposure circumstances that have been obtained over the years in the chrysotile asbestos cement manufacturing factories in Zimbabwe.

## 5. Conclusions

The results suggest that saw cutting operators, followed by kollergang and ground hard waste operators, in both factories may be at increased risk of developing ARDs, as a high number of predicted cancer mortality cases and asbestosis cases, especially if workers were exposed before 2008, where exposure concentrations and subsequently cumulative exposure, would be high compared to exposure concentrations for the time period of 2009 to 2020, assuming that workers continued to be exposed for more than 20 years from around the 1990s, as regular chrysotile exposure data began to be available in the early to mid-1990s. A duration of 20 years or more is commonly associated with a long latency period of ARDs [1,2].

Furthermore, ARDs predicted in this study, although they were generally lower than reported in other studies that used the OSHA risk assessment model, underline the need for asbestos mortality and or morbidity studies in Zimbabwe with the aim of a comprehensive inquiry of a comparative analysis of predicted and actual mortality cases that may be obtained in the AC manufacturing industry in Zimbabwe. It is essential that mortality and/or morbidity epidemiology studies be further carried out to establish the possible actual disease burden associated with exposure to chrysotile asbestos in the AC manufacturing factories in Zimbabwe.

## Figures and Tables

**Figure 1 ijerph-20-00058-f001:**
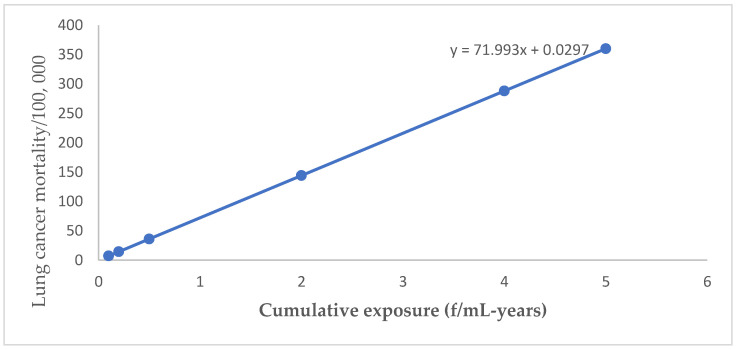
Lung cancer mortality by cumulative exposure linear regression equation.

**Figure 2 ijerph-20-00058-f002:**
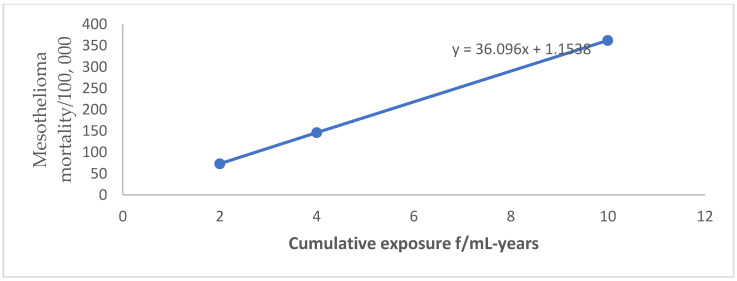
Mesothelioma mortality by cumulative exposure linear regression equation.

**Figure 3 ijerph-20-00058-f003:**
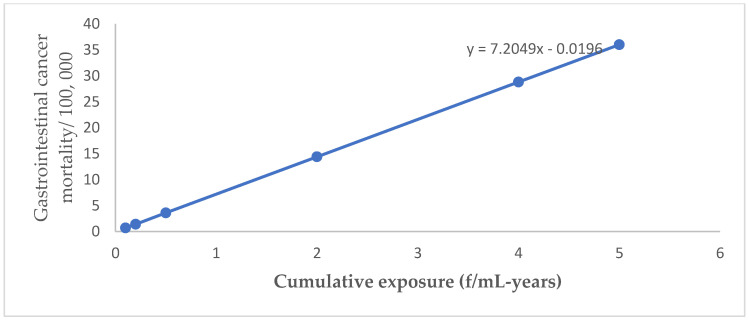
Gastrointestinal cancer mortality by cumulative exposure linear regression equation.

**Table 1 ijerph-20-00058-t001:** Estimates of asbestos-related cancer mortality per 100,000 exposed workers by number of years exposed and exposure level.

Asbestos Fibre Concentration (f/mL)	Cancer Mortality per 100,000 Exposed Workers			
	Lung	Mesothelioma	Gastrointestinal	Total
1-year exposure				
0.1	7.2	6.9	0.7	14.8
0.2	14,4	13.8	1.4	29.6
0.5	36.1	34.6	3.6	74.3
2.0	144	138	14.4	296.4
4.0	288	275	28.8	591.8
5.0	360	344	36.0	740.0
10.0	715	684	71.5	1470.5
20-year exposure				
0.1	139	73	13.9	225.9
0.2	278	146	27.8	451.8
0.5	692	362	69.2	1123.2
2.0	2713	1408	271.3	4392.3
4.0	5278	2706	527.8	8511.8
5.0	6509	3317	650.9	10,476.9
10.0	12,177	6024	1217.7	13,996.7
45-year exposure				
0.1	231	82	23.1	336.1
0.2	460	164	46.0	670.0
0.5	1143	407	114.3	1664.3
2.0	4416	1554	441.6	6411.6
4.0	8441	2924	844.1	12,209.1
5.0	10,318	3547	1031.8	14,896.8
10.0	18,515	6141	1851.5	26,507.5

Source: OSHA, 1986; MSHA, 2005 and 2008.

**Table 2 ijerph-20-00058-t002:** Abridged job-exposure matrix based on mean airborne chrysotile exposure data per time-period and place for the period of 1996–2020.

Job	Job Description	Time Period	Harare Factory	Bulawayo Factory
			Mean (f/mL)	Mean (f/mL)
Saw cutting operator (1023)	Cutting by saw asbestos sheets and facia boards to size	1996–2000	0.19 ± 0.01	0.17 ± 0.02
		2001–2008	0.13 ± 0.02	0.12 ± 0.02
		2009–2016	0.07 ± 0.02	0.06 ± 0.02
		2018–2020 *	0.10 ± 0.02	0.05 ± 0.01
Fettling table operator	Scrapping off small protrusions and polishing AC moulded goods to make them smooth	1996–2000	0.12 ± 0.04	0.17 ± 0.06
		2001–2008	0.12 ± 0.02	0.12 ± 0.03
		2009–2016	-	-
		2018–2020 *	0.11 ± 0.03	-
Moulded goods operator	Moulding of AC goods under wet conditions	1996–2000	0.11 ± 0.04	-
		2001–2008	0.11 ± 0.04	-
		2009–2016	0.05 ± 0.01	-
		2018–2020 *	0.11 ± 0.02	-
Kollergang operator	Opening of and loading chrysotile bags into mouth of process machine and operating machine to move chrysotile into production line	1996–2000	0.13 ± 0.04	0.14 ± 0.03
		2001–2008	0.12 ± 0.02	0.12 ± 0.01
		2009–2016	0.07 ± 0.02	0.07 ± 0.03
		2018–2020 *	0.12 ± 0.01	0.06 ± 0.01
Ground hard waste operator	Feeding AC waste materials into grinder machine	1996–2000	0.16 ± 0.03	0.13 ± 0.04
		2001–2008	0.13 ± 0.03	0.11 ± 0.04
		2009–2016	0.07 ± 0.02	0.07 ± 0.02
		2018–2020 *	0.12 ± 0.01	0.06 ± 0.02
Laundry room operator	Laundering of PPC using wash machine	1996–2000	0.13 ± 0.03	-
		2001–2008	0.13 ± 0.02	-
		2009–2016	0.05 ± 0.01	-
		2018–2020 *	0.11 ± 0.02	-
Pipe joints operator	Lathe machining of AC joints pipes	1996–2000	-	0.13 ± 0.04
		2001–2008	-	0.11 ± 0.01
		2009–2016	-	0.05 ± 0.02
		2018–2020	-	0.05 ± 0.02
Full-length pipe operator	Lathe machining and polishing of full-length AC pipe joints	1996–2000	-	0.13 ± 0.04
		2001–2008	-	0.11 ± 0.01
		2009–2016	-	0.07 ± 0.02
		2018–2020	-	-
Multi-cutter operator	Cutting full-length pipes into collars for coupling pipes	1996–2000	-	0.13 ± 0.04
		2001–2008	-	0.12 ± 0.01
		2009–2016	-	0.07 ± 0.03
		2018–2020	-	0.04 ± 0.01

Source: Mutetwa et al., 2022 [33]; * Care and maintenance of equipment and cleaning.

**Table 3 ijerph-20-00058-t003:** Estimates of the summary of total cancer mortality cases by job and time period.

		Harare Factory					Bulawayo Factory				
			Total Cancer Cases per 100,000 Exposed/(% Incidence)					Total Cancer Cases per 100,000 Exposed/(% Incidence)			
Job	Time Period	Mean (f/mL)	1 Year	10 Years	20 Years	25 Years	Mean (f/mL)	1 Year	10 Years	20 Years	25 Years
Saw cutting operator	1996–2000	0.19	23 (0.02)	219 (0.22)	439 (0.44)	555 (0.56)	0.17	21 (0.02)	197 (0.20)	394 (0.39)	497 (0.50)
	2001–2008	0.13	16 (0.02)	151 (0.15)	301 (0.30)	382 (0.38)	0.12	15 (0.02)	137 (0.14)	277 (0.28)	347 (0.35)
	2009–2016	0.07	9 (0.01)	81 (0.08)	163 (0.16)	209 (0.21)	0.06	8 (0.01)	63 (0.06)	137 (0.14)	174 (0.17)
	2018–2020	0.10	13 (0.01)	116 (0.12)	231 (0.23)	289 (0.29)	0.05	7 (0.01)	59 (0.06)	116 (0.12)	151 (0.15)
Fettling table operator	1996–2000	0.12	15 (0.02)	137 (0.14)	277 (0.28)	347 (0.35)	0.17	21 (0.02)	197 (0.20)	394 (0.39)	497 (0.56)
	2001–2008	0.12	15 (0.02)	137 (0.14)	277 (0.28)	347 (0.35)	0.12	15 (0.02)	137 (0.14)	277 (0.28)	347 (0.40)
	2009–2016	-	-	-	-	-	-	-	-	-	-
	2018–2020	0.11	14 (0.01)	128 (0.13)	255 (0.26)	324 (0.32)	-	-	-	-	-
Moulded goods operator	1996–2000	0.11	14 (0.01)	128 (0.13)	255 (0.26)	324 (0.32)	-	-	-	-	-
	2001–2008	0.11	14 (0.01)	128 (0.13)	255 (0.26)	324 (0.32)	-	-	-	-	-
	2009–2016	0.05	7 (0.01)	59 (0.06)	116 (0.12)	151 (0.15)	-	-	-	-	-
	2018–2020	0.11	14 (0.01)	128 (0.13)	255 (0.26)	324 (0.32)	-	-	-	-	-
Kollergang operator	1996–2000	0.13	16 (0.02)	151 (0.15)	301 (0.30)	381 (0.38)	0.14	17 (0.02)	163 (0.16)	324 (0.32)	404 (0.40)
	2001–2008	0.12	15 (0.02)	137 (0.14)	277 (0.28)	347 (0.35)	0.12	15 (0.02)	137 (0.14)	277 (0.28)	347 (0.35)
	2009–2016	0.07	9 (0.01)	81 (0.08)	163 (0.16)	209 (0.21)	0.07	9 (0.01)	81 (0.08)	163 (0.16)	209 (0.21)
	2018–2020	0.12	15 (0.02)	137 (0.14)	277 (0.28)	347 (0.35)	0.06	8 (0.01)	63 (0.06)	137 (0.14)	174 (0.17)
Ground hard waste operator	1996–2000	0.16	20 (0.02)	186 (0.19)	370 (0.37)	463 (0.46)	0.13	16 (0.02)	151 (0.15)	301 (0.30)	381 (0.38)
	2001–2008	0.13	16 (0.02)	151 (0.15)	301 (0.30)	381 (0.38)	0.11	14 (0.01)	128 (0.13)	255 (0.26)	324 (0.32)
	2009–2016	0.07	9 (0.01)	81 (0.08)	163 (0.16)	209 (0.21)	0.07	9 (0.01)	81 (0.08)	163 (0.16)	209 (0.21)
	2018–2020	0.12	15 (0.02)	137 (0.14)	277 (0.28)	347 (0.35)	0.06	8 (0.01)	63 (0.06)	137 (0.14)	174 (0.17)
Laundry room operator	1996–2000	0.13	16 (0.02)	151(0.15)	301 (0.30)	382(0.38)	-	-	-	-	-
	2001–2008	0.13	16 (0.02)	151(0.15)	301 (0.30)	382 (0.38)	-	-	-	-	-
	2009–2016	0.05	7 (0.01)	59 (0.06)	116 (0.12)	151 (0.15)	-	-	-	-	-
	2018–2020	0.11	14 (0.01)	128(0.13)	255 (0.26)	324 (0.32)	-	-	-	-	-
Pipe joints operators	1996–2000	-	-	-	-	-	0.13	16 (0.02)	151 (0.15)	301 (0.30)	381 (0.38)
	2001–2008	-	-	-	-	-	0.11	14 0.01)	128 (0.13)	255 (0.26)	324 (0.32)
	2009–2016	-	-	-	-	-	0.05	7 (0.01)	59 (0.06)	116 (0.12)	151 (0.15)
	2018–2020	-	-	-	-	-	0.05	7 (0.01)	59 (0.06)	116 (0.12)	151 (0.15)
Full-length pipe operator	1996–2000	-	-	-	-	-	0.13	16 (0.02)	151 (0.15)	301 (0.30)	381 (0.38)
	2001–2008	-	-	-	-	-	0.11	14 (0.01)	128 (0.13)	255 (0.26)	324 (0.32)
	2009–2016	-	-	-	-	-	0.07	9 (0.01)	81 (0.08)	163 (0.16)	209 (0.21)
	2018–2020	-	-	-	-	-	-	-	-	-	-
Multi-cutter operator	1996–2000	-	-	-	-	-	0.13	16 (0.02)	151 (0.15)	301 (0.30)	381 (0.38)
	2001–2008	-	-	-	-	-	0.12	15 (0.02)	137 (0.14)	277 (0.28)	347 (0.34)
	2009–2016	-	-	-	-	-	0.07	9 (0.01)	81 (0.08)	163 (0.16)	209 (0.21)
	2018–2020	-	-	-	-	-	0.04	6 (0.01)	48 0.05)	105 (0.10)	116 (0.12)
Overall factory		0.11	14 (0.01)	128 (0.13)	255 (0.26)	324 (0.32)	0.12	15 (0.02)	137 (0.14)	277 (0.28)	347 (0.35)

The predicted cancer cases highlighted in Table 3 are based on Appendix A. For instance, assuming that exposures associated with the time period of 1996 to 2000 for saw cutting operator in the Harare factory on average persists for 1 year, Appendix A (lung cancer) shows 13.7 cases per 100,000, Appendix A (mesothelioma) shows 8 cases, and Appendix A (gastrointestinal cancer) shows 1.3 cases per 100,000 workers. These cases added, i.e., 13.7 + 8 + 1.3, give 23 cases per 100,000. This process was repeated for all operators across all exposure periods of 1, 10, 20, and 25 years to come up with the summary for Table 3.

**Table 4 ijerph-20-00058-t004:** Cumulative exposure and asbestosis incidence by time period and after 25 years of exposure.

		Harare Factory				Bulawayo Factory			
Job	Time Period	Mean Concentration (f/mL)	CE	% Incid	Case/ 100 × 10^3^	Mean Concentration (f/mL)	CE	% Incid	Case/ 100 × 10^3^
Saw cutting operator	1996–2000	0.19	4.8	0.26	260	0.17	4.3	0.24	240
	2001–2008	0.13	3.3	0.18	180	0.12	3.0	0.17	170
	2009–2016	0.07	1.8	0.10	100	0.06	1.5	0.08	80
	2018–2020	0.10	2.5	0.14	140	0.05	1.3	0.07	70
Fettling table operator	1996–2000	0.12	3.0	0.17	170	0.17	4.3	0.24	240
	2001–2008	0.12	3.0	0.17	170	0.12	3.0	0.17	170
	2009–2016	-	-	-	-	-	-	-	-
	2018–2020	0.11	2.8	0.15	150	-	-	-	-
Moulded goods operator	1996–2000	0.11	2.8	0.15	150	-	-	-	-
	2001–2008	0.11	2.8	0.15	150	-	-	-	-
	2009–2016	0.05	1.3	0.07	70	-	-	-	-
	2018–2020	0.11	2.8	0.15	150	-	-	-	-
Kollergang operator	1996–2000	0.13	3.3	0.18	180	0.14	3.5	0.19	190
	2001–2008	0.12	3.0	0.17	180	0.12	3.0	0.17	170
	2009–2016	0.07	1.8	0.10	100	0.07	1.8	0.10	100
	2018–2020	0.12	3.0	0.17	170	0.06	1.5	0.08	80
Ground hard waste operator	1996–2000	0.16	4.0	0.22	220	0.13	3.3	0.18	180
	2001–2008	0.13	3.3	0.18	180	0.11	2.8	0.15	150
	2009–2016	0.07	1.8	0.10	100	0.07	1.8	0.10	100
	2018–2020	0.12	3.0	0.17	170	0.06	1.5	0.08	80
Laundry room operator	1996–2000	0.13	3.3	0.18	180	-	-	-	-
	2001–2008	0.13	3.3	0.18	180	-	-	-	-
	2009–2016	0.05	1.3	0.07	70	-	-	-	-
	2018–2020	0.11	2.8	0.15	150	-	-	-	-
Pipe joints operator	1996–2000	-	-	-	-	0.13	3.3	0.18	180
	2001–2008	-	-	-	-	0.11	2.8	0.15	150
	2009–2016	-	-	-	-	0.05	1.3	0.07	70
	2018–2020	-	-	-	-	0.05	1.3	0.07	70
Full-length pipe operator	1996–2000	-	-	-	-	0.13	3.3	0.18	180
	2001–2008	-	-	-	-	0.11	2.8	0.15	150
	2009–2016	-	-	-	-	0.07	1.8	0.10	100
	2018–2020	-	-	-	-	-	-	-	-
Multi-cutter operator	1996–2000	-	-	-	-	0.13	3.3	0.18	180
	2001–2008	-	-	-	-	0.12	3.0	0.17	300
	2009–2016	-	-	-	-	0.07	1.8	0.10	100
	2018–2020	-	-	-	-	0.04	1.0	0.05	50
Overall factory		0.11	2.8	0.15	150	0.12	3.0	0.17	170

% Incid—percentage incidence; CE—cumulative exposure in f/mL-years.

## Data Availability

The dataset used in this study are available from the corresponding author on reasonable request. The datasets are not publicly available to maintain confidentiality of the factories used in the study.

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
