# Peer review of "Prediction of Asbestos-Related Diseases (ARDs) and Chrysotile Asbestos Exposure Concentrations in Asbestos-Cement (AC) Manufacturing Factories in Zimbabwe"

_ijerph, 2022, doi:10.3390/ijerph20010058_

Round 1

Reviewer 1 Report

Introduction

·       Occupational exposure to asbestos is associated with asbestos related diseases  (ARDs) namely asbestosis, lung cancer and mesothelioma, a rare form of malignant tumour of the lining of the chest or abdominal cavities. THIS SENTENCE IS UNCOMPLEATE SINCE IT IS LIMITED TO SOME FORMS OF ARDs ONLY.

·       Other health outcomes associated with asbestos exposure include cancer of pharynx, cancer of the oesophagus, cancer of the larynx and cancer of the stomach. EVEN THIS SENTENCE IS NOT COMPLETE NOR CORRECT, SINCE IT MIXES CANCERs WITH DIFFERENT EVIDENCE (laryinx and oesophagus) AND LACKS OF SOME FORMS, SUCH AS OVARIC CANCER

·       chrysotile induced asbestosis in so far as increasing levels…SENTENCE NOT CLEAR

·       Reduction of exposure to 0.2 f/ml fibre concentration was reported to result in a life- time incidence of asbestos of about 0.5%. PLEASE CORRECT ASBESTOS

·       INTRODUCTION MUST BE SHORTENED

Methods

·      Clear

Results

·      It was hard for me to go into table details. Tables are too many and too stuffed with data. There are 12 tables in suppl materials!!! Too many

Discussion

OSHA [23], has always 365 considered that, risk of mortality of more than 1 case per 1000 workers from occupational 366 causes presents a significant risk. IS THIS VALID ALSO FOR OTHER INSTITUTIONS?

Author Response

Dear Reviewer

Thank you for taking time to review our manuscript. Attached is our responses to the comments.

Reviewer 2 Report

I read with a great interest this manuscript dealing with predictions of cancer cases and asbestosis among two different group of workers of differente AC factories in Zimbabwe. 

I think that this could be an interesting paper suitable to improve awerness of absestos related diseases and their prevention, once published. I have only two major observations:

1) Authors carried out their analyses on mesotheliomas, lung cancer and gastrointestinal cancers: to my knowledge no clear and definitive relationship was underlined yet, considering asbestos exposure and gastrointestinal cancers. On the other hand, larynx and ovary cancer are causally related to asbestos exposure following IARC classification. I think that Authors should explain this choice and discuss it.

2) The model used is linear, although some papers pointed out that, at least for pleural mesothelioma, the relationship between exposure and number of cases could increased almost linearly for lower levels of exposure but it could flattened at higher levels. In my opinion, Authors should discuss more extensively how this could affect their estimates. Moreover, there could be differences in the shape of risk function considering pleural or peritoneal mesothelioma. If Authors considered only pleural mesothelioma, it should stressed within the text and also in the tables.

Author Response

Thank you for taking time to review our manuscript. Attached is our responses to the comments.

Reviewer 3 Report

General comments The paper and particularly the discussion would benefit from a language revision and shortening.  It is not generally accepted that there is a risk of asbestosis at cumulative levels as low as those reported (though I personally agree the this risk exists even at these levels). If the OSHA report on which the linear model is based has any documentation it would probably be beneficial to state it shortly.     Specific comments:  

Abstract 

l.27 mentions "saw cutting, kollergang, fettling table, ground hard waste and possibly pipe making operations” whereas in the discussion you emphasize only "saw cutting operators followed by kollergang and ground 296 hard waste operators” The abstract should be aligned with the discussion. l. 33 How can the study inform mortality studies as you claim, when it only includes incidence outcomes?

Author Response

Thank you for taking time to review our manuscript. Attached is our responses to the comments

Reviewer 4 Report

Line 212: Although the topic of this paper fits the scope of the journal, the assessment methodology applied in this paper lacks novelty and is very ordinary. Consider using more advanced methodologies for analysis such as machine learning and artificial intelligence.

Line 221: Does not comply with the format of the journal. The Ethics section should appear at the end of the paper, not within the text.

Line 409: Does not comply with the format of the journal. Please refer to the journal template and consider removing Section 5. Strengths and Weaknesses.

The paper currently is not up to journal publication standards. The paper must undergo major revision before consideration for publication.

Author Response

Thank you for taking time to review our manuscript, Attached is our responses to the comments

Reviewer 5 Report

This is a study on the predict possible Asbestos Related Diseases (ARDs) namely lung cancer, mesothelioma, gastrointestinal cancer and asbestosis, in two chrysotile asbestos cement (AC) manufacturing factories.

The authors demonstrated that ARDs namely lung cancer, mesothelioma, gastrointestinal cancer and asbestosis may be anticipated at various cumulative exposures over 1, 10, 20, and 25 years of exposure in AC manufacturing factories in Zimbabwe.

This is a well-written paper, and would provide some useful information on the ARDs correlation with chrysotile asbestos exposure concentrations to readers. I have a few minor comments.

1. Why did the exposure levels and subsequently cumulative exposure increase after 2008?

2. Line 275. Table 4, Correct the font in the same shape.

3. Line 696. Table S8, Correct the font in the same shape.

4. Line 703. Table S9, Correct the font in the same shape.

5. Line 709. Table S10, Correct the font in the same shape.

Author Response

(The authors gave the same response as above.)

Round 2

Reviewer 1 Report

Authors addressed properly the points I raised

Author Response

Thank you for taking time from busy schedule to review our manuscript

Regards

Benjamin Mutetwa

Reviewer 2 Report

I think that the manuscript was improved by Authors and they correctly answered to my suggestions.
I think that manuscript could be published in the present form.

Author Response

Thank you for taking time from your busy schedule to review our manuscript.

Regards

Benjamin Mutetwa

Reviewer 4 Report

Although the authors did not give their best efforts to improve the manuscript, I have no objection for the manuscript to be published in present form.

Author Response

Thank you for taking time to review our manuscript.

Regards

Benjamin Mutetwa
